# Extending the life of wind turbine blade leading edges by reducing the tip speed during extreme precipitation events

Jakob I. Bech[1], Charlotte B. Hasager[1], Christian Bak[1]

[1]Department of Wind Energy, Technical University of Denmark, Roskilde, 4000, Denmark

5  *Correspondence to*: Jakob Ilsted Bech (jakb@dtu.dk)

**Abstract.** Impact fatigue caused by collision with rain droplets, hail stones and other airborne particles, also known as leading edge erosion, is a severe problem for wind turbine blades. Each impact on the leading edge adds an increment to the accumulated damage in the material. After a number of impacts the leading edge material will crack. This paper presents and supports the hypothesis that the vast majority of the damage accumulated in the leading edge is imposed at extreme

10  precipitation condition events, which occur during a very small fraction of the turbines operation life. By reducing the tip speed of the blades during these events, the service life of the leading edges significantly increases from a few years to the full expected lifetime of the wind turbine. This life extension may cost a negligible reduction of annual energy production (AEP) in the worst case, and in the best case a significant increase in AEP will be achieved.

**Nomenclature and abbreviations**

| | | |
|---|---|---|
| *AEP* | annual energy production | |
| *BEM* | Blade element momentum | |
| *AOA* | angle of attack | [rad] |
| *CI* | cost of inspection per rotor | [€] |
| *CR* | cost of repair per rotor | [€] |
| *DTV* | camage threshold velocity | [m/s] |
| *EC* | energy cost | [€/kWh] |
| *ECS* | erosion control strategy | |
| *GFRP* | glas fiber reinforced polymer | |
| *LEE* | leading edge erosion | |
| *LEP* | leading edge protection | |
| *LER* | leading edge roughness | [m] |
| *NOI* | number of inspections per rotor | [-] |
| *NOR* | number of repairs per rotor | [-] |
| *TEP* | Total energy production | [kWh] |
| *WA-RET* | whirling arm rain eerosion tester | |
| *A* | Weibull parameter | [m/s] |
| *C* | Weibull parameter | [-] |
| *c* | constant | [-] |

| $c_c$ | airfoil chord length | [m] |
|---|---|---|
| $c_d$ | drag coefficient | [-] |
| $c_l$ | lift coefficient | [-] |
| $c_l$ | speed of sound in liquid | [m/s] |
| $c_s$ | speed of Rayleigh wave in target material | [m/s] |
| $D$ | droplet diameter | [m] |
| $E_0$ | 1 J | [J] |
| $E_k$ | kinetic energy, of each impact of droplet | [J] |
| $F$ | specific impact frequency - impacts per area per time | $[s^{-1}m^{-2}]$ |
| $I_r$ | rain intensity | [m/s] |
| $m$ | exponent of Wöhler curve | [-] |
| $m$ | mass | [kg] |
| $M$ | Miner sum | [-] |
| $N$ | number of droplets per volume | $[m^{-3}]$ |
| $N_{Ei}$ | number of impacts per unit area to removal of coating | $[m^{-2}]$ |
| $N_i$ | number of cycles to fatigue failure | [-] |
| $n_i$ | number of cycles | [-] |
| $P_0$ | rated mechanical power with original control | [W] |
| $P_1$ | rated mechanical power with erosion control | [W] |
| $Q_0$ | original rated main shaft torque | [Nm] |
| $t$ | airfoil thickness | [m] |
| $t_i$ | test time to removal of coating | [s] |
| $V$ | relative volume of water in rain field | [-] |
| $v$ | impact velocity | [m/s] |
| $v_r$ | droplet falling velocity | [m/s] |
| $\rho_l$ | specific density of liquid | $[kg/m^3]$ |
| $\rho_s$ | specific density of solid | $[kg/m^3]$ |
| $\omega_0$ | maximum rotational speed with original control | [rad/s] |
| $\omega_1$ | maximum rotational speed with erosion control | [rad/s] |

## 1 Introduction

Leading edge erosion (LEE) is a severe problem for the wind energy sector today (Keegan et al., 2013; Slot et al., 2015). Wind turbine operators report significant costs for inspection, maintenance, repair, and loss of energy production due to down time and reduced performance (Stephenson, 2011). LEE increases the surface roughness of blades and deteriorates the aerodynamic performance resulting in lower annual energy production (AEP) during turbine operation (Zidane et al., 2016). The LEE issue

has appeared as a consequence of the trend towards larger turbines with longer blades and higher nominal tip speeds (Keegan et al., 2013; Macdonald et al., 2016). As an example, recently 273 blades with less than 7 years in operation were refurbished at an offshore wind farm in the North Sea. Some of the blades were even removed and taken ashore for repair of damages due to LEE (Wittrup, 2015). During the review phase of this paper, it has been revealed, that several blades of 111 3.6 MW turbines at the Anholt offshore wind farm will be dismantled and brought ashore for repair of leading edge erosion damage less than 5 years after it was inaugurated. Similar repair campaigns are foreseen for the London array with 175 similar turbines and other UK offshore wind farms (Renews 2018a; Renews 2018b; OffshoreWind.Biz 2018).

LEE is caused by a multitude of factors within the atmospheric environment and the leading edge structure. In addition to rain, impacts of sand particles (Zidane et al., 2017) and other airborne particles such as hail stones (Macdonald et al., 2015) and insects, global strain from blade flexing, temperature oscillations, UV radiation and long term exposure to moisture, chemicals and salt also add to the material degradation. Efforts to understand rain-induced erosion include simulation (Blowers, 1969; Springer, 1975; Sloth et al., 2015; Amirzadeh, 2017) and laboratory testing (Bowden et al., 1964; Keegan et al., 2013). A thorough understanding of rain erosion of layered anisotropic polymer-based-structures like wind turbine blades is not yet available. However, it is clear, that several damage mechanisms are observed, and that the impact velocity is a governing factor as well as the amount of precipitation and the structure and materials of the leading edge (Siddons et al., 2015; Cortés et al., 2017).

The industrial standard for measuring the durability of leading edge strucures is the whirling arm rain erosion test (WA-RET) (ASTM G73-10, 2010, Liersch 2014, DNVGL-RP-0171). In the WA-RET the test specimens are mounted on a rotor spinning at high velocity in an artificially generated rain field. The rotor velocity, rain intensity and droplet size are carefully controlled, as impact velocity, droplet dimension and number of impacts are the governing factors for the magnitude of damage imposed on a given test specimen (Adler, 1999). It should be kept in mind, that the whirling arm test method does not reflect the real operating conditions for rain impact. The impact velocities of the accelerated tests are typically up to two times the tip speed of real blades, which may cause irrelevant failure modes. Also the fixed rain field with constant rain intensity and drop size distribution is very different from field conditions, where droplet sizes and rain intensity vary a lot (Best 1950).

Blade- and turbine manufacturers as well as coating suppliers put effort to develop and implement leading edge protection structures that will last the expected lifetime of the turbines. Wind turbine (WT) operators put effort to define feasible inspection and service intervals and to repair damaged blades. The latest developments in leading edge protection (LEP) applied to new turbines have yet to prove their durability in long term field conditions. Already installed turbines without the latest inventions in LEP are still vulnerable to erosion, and repairs made on site may have varying quality. Also, in order to reduce the torques and loads, it may be attractive to increase the tip speeds even further on future turbine designs. Consequently, alternative strategies of mitigation of LEE should be explored.

Such an alternative strategy is the reduction of the tip speed during highly erosive conditions (Wobben, 2003). It is likely to be feasible to extend the leading edge life by reducing the rotor speed during extreme precipitation events occurring at a very

little fraction of the service life, but accounting for the majority of the erosion damage. The threshold values of precipitation as indicator for tip speed reduction will be determined for the individual wind turbine plant as described in section 5. The approach to erosion control is inspired by aerodynamic load control, where it is a common strategy to reduce the extreme loads caused by gusts and turbulence by pitching out the blades under these conditions. Such systems operate automatically in modern wind turbines. (Njiri et al., 2016).

The objective of this paper is to present and support the hypothesis on mitigation of leading edge erosion by control of wind turbines during high intensity rain events. In section 2 some important aspects of leading edge erosion are presented with focus on liquid droplet impact stresses and fatigue. Section 3 presents an analysis of whirling arm rain erosion test data provided by Polytech A/S. The analysis includes introduction to block loading and a cumulative damage law. Section 4 presents precipitation parameters and their statistical occurrence, while section 5 focuses on turbine control for reducing tip speed and includes control strategies with different loss of production vs. extension of life of blades. The discussion follows in section 6 and conclusions are drawn in section 7.

## 2 Rain erosion of leading edge

### 2.1 Droplet impact

Rain erosion is the consequence of multiple impacts stochastically distributed over the surface of the coated laminate. Each impact adds a damage increment to the accumulated damage. For rain and other air borne particles the accumulated damage is a function of several parameters including the number of impacts per unit area and the magnitude of each impact. This paper is limited to consider impact by liquid droplets only.

The magnitude of an impact of a droplet hitting perpendicular to the surface may be quantified by the kinetic energy ($E_{kin}$)

$$E_k = \frac{1}{2}mv^2 \tag{1}$$

where $v$ is the velocity of the particle relative to the surface and $m$ is the mass of the droplet.

For detailed analysis the impact may be quantified by the contact stress field acting on the surface as a function of time during the impact (Keegan et al., 2012). The contact stresses are functions of the properties of the liquid, the properties of the impacted surface, the impact velocity and the size and shape of the droplet.

The impact of a spherical droplet immediately causes a normal pressure on the target surface at the initial point of contact. The contact area between the droplet and the solid expands radially at a velocity higher than the speed of sound in water. When the shock wave front reaches the edge of the droplet, a release jet is generated, and the pressure reduces to the stagnation pressure (Bowden 1964, Dear and Field, 1988).

The simplest expression for calculation of the initial contact pressure is the water hammer equation (Bowden, 1961). It was derived for a column of liquid impacting a rigid surface, where a compression wave propagates from the contact into the liquid. The immediate contact pressure ($p$) may be calculated by

$$p = v\rho_l c_l \tag{2}$$

where $v$ is the impact velocity, $c_l$ is the speed of the compression wave in the liquid, and $\rho_l$ is the density of the liquid. Accounting for the geometry of a spherical droplet, the contact angle increases as the contact area expands, and the peak pressure at the rim of the contact is analytically derived (Heymann, 1969) as

$$p = 3v\rho_l c_l \tag{3}$$

Taking into account the compliance of the solid, the pressure of impact between an elastic solid cylinder and a liquid jet (de Haller 1933) may be expressed as

$$p = v\frac{\rho_l c_l \rho_s c_s}{\rho_l c_l + \rho_s c_s} \tag{4}$$

where sub-script $l$ is for liquid and $s$ for solid. Later numerical modelling works take into account an assumed spherical
geometry of the droplet as well as the compliance of the target material (Adler 1995, Amirzadeh et al., 2017). These studies also show a pressure peak near the edge of the contact.

Real precipitation droplets falling through the atmosphere are not necessarily spherical. The aerodynamic forces distort the droplet to a burger bun-like shape. Larger droplets, d>6 mm, flatten out before splitting up (Fakhari 2009), while smaller droplets tend to merge and form larger droplets. The droplet geometry may be characterized by its ratio of vertical to horizontal
dimensions (Gorgucci et al., 2006). Through a full rotation of 360 degrees the wind turbine blades are hitting the non-spherical droplets from all angles at different relative velocities. This makes the impact scenario even more complex.

### 2.2 Impact stresses, fracture and fatigue

A typical leading edge consists of a curved laminate of Glass Fiber-Reinforced Polymer (GFRP) with a relatively brittle polyurethane, polyester or epoxy –based coating. Many designs have a layer of putty or filler, applied to the laminate and
sanded, to make a smooth surface for the coating. Recent developments have added a top layer of elastomeric coating with good damping properties and high fracture toughness, often referred to as leading edge protection or LEP, see Fig. 1, (Cortés et al., 2017).

An impact on the surface causes stress transients in the material. Stress waves propagate from the impact site into the coated composite (body waves) and along its surface (surface waves). Several stress components are active as functions of the time
after the impact, the radial planar position and depth in the material (Woods 1968, Adler 1995). These stresses can activate different failure modes depending on the velocity of the impact, the size of the droplet and "the weakest link" in the leading edge structure.

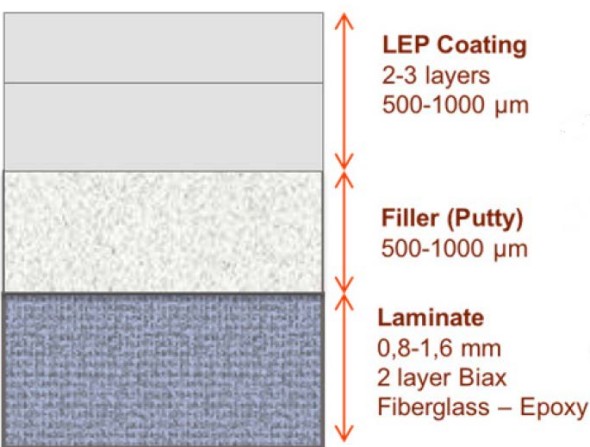

**Figure 1: Example of leading edge protection system configuration (Cortés et al., 2017).**

For isotropic, homogenous, elastic materials ring shaped surface cracks due to Rayleigh waves is the common type of impact damage (Blowers 1969, Bowden, 1964). For many coated materials, this may also be the governing failure mechanism.

The body waves propagating perpendicular to the surface into the target material can lead to sub-surface cracks in the coating and the substrate (Fraisse et al., 2018). The reflection of stress waves between the coating surface and the coating/substrate interface may also play a significant role for fatigue of the coated laminate (Springer 1974). Body waves may also cause delamination inside the laminate (Prayago 2011) and debonding of the coating (Cortés et al., 2017).

A single droplet impact may cause instant damage, when the impact velocity is beyond the damage threshold velocity (DTV). For a given droplet size and set of material parameters, DTV was derived for brittle materials by a fracture mechanics approach, (Evans et al., 1980).

$$DTV = v_p^c = \sqrt[3]{\lambda \frac{2K_c^2 c_s}{D\rho_l^2 c_l^2}} \qquad (5)$$

where $K_c$ is the critical stress intensity factor and $c_s$ is the Rayleigh wave velocity of the target material, $D$ is the diameter of the spherical droplet, $\rho_l$ is the density of the liquid, $c_l$ is the speed of sound in the liquid, and $\lambda$ is a material independent constant.

Repeated stresses below the static strength of a material may eventually cause failure due to cumulative fatigue damage (Minor 1945). Similarly, impacts below DTV may also add to the accumulated damage, which may eventually cause fracture (Springer 1975). For materials with a fatigue limit, like some metals, a fatigue threshold impact velocity, below which no erosion will occur, may be defined as an analogy to the endurance limit found in fatigue testing (Heymann, 1969). Rain erosion test data can be regarded as impact fatigue data. Together with operational data for a wind turbine and the local precipitation statistics it can be used to predict the erosion propagation and lifetime of leading edges in field operation (Eisenberg et al., 2016).

# 3 Empirical rain erosion test data

## 3.1 Analysis of rain erosion test data

An example of rain erosion test data was made available by Polytech A/S, see Fig. 2. The test specimen material is coated aluminum. The specimen has a length of 225 mm. In this test, the tangential velocity was 140 m/s at the tip and 110 m/s at the root of the specimen. The rain intensity was 30-35 mm/h and droplet sizes were ranging from 1 to 2 mm. (In the later calculations, for simplicity, it is assumed, that the rain intensity is 32.5 mm/h, and the droplet diameter is uniform at 2 mm. The falling velocity of the droplets is assumed to be 6 m/s, corresponding to the terminal velocity of 2 mm droplets (Foote et al., 1969). The test was stopped every 30 minutes for photography of the specimens, see Fig. 2. The photographs are used to determine the progression of erosion. Here erosion is defined as visible removal of the top coat. The erosion initiates at the tip of the specimen, where velocity is highest. It then propagates towards the root, where the velocity is lower. Each position on the specimen corresponds to a certain tangential velocity. The data pairs of the position of the erosion front and the time are shown in table 1 along with the corresponding local rotor velocities. The kinetic energy of each impact and the number of impacts per unit area are explained in section 3.2.

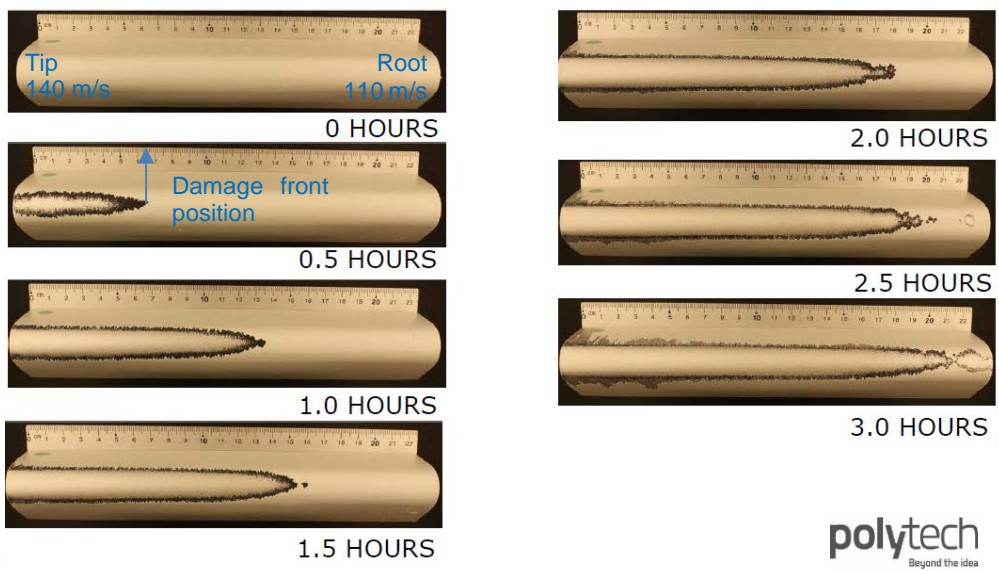

Figure 2: Whirling arm rain erosion test specimen photographed at 30 minutes intervals during 3 hours of testing.

**Table 1: Erosion propagation as function of time**

| Test time [hrs] | Position of erosion front [mm] | Accumulated rain [mm] | Tangential speed at erosion front [m/s] | Kinetic energy, of each impact [J ] | Impacts/area to removed coating [cm$^{-2}$] |
|---|---|---|---|---|---|
| 0.5 | 66 | 16 | 131 | 0.036 | 8.5E+03 |
| 1.0 | 135 | 33 | 122 | 0.031 | 1.6E+04 |
| 1.5 | 161 | 49 | 119 | 0.029 | 2.3E+04 |
| 2.0 | 182 | 65 | 116 | 0.028 | 3.0E+04 |
| 2.5 | 208 | 81 | 112 | 0.026 | 3.6E+04 |
| 3.0 | 228 | 98 | 110 | 0.025 | 4.3E+04 |

The data on time to removed coating as function of the local rotor speed from table 1 are plotted in Fig. 3

Rain erosion can be analyzed as a fatigue process (Slot et al., 2015), where, traditionally, the independent parameter, here

5  velocity, is plotted on the vertical axis and the dependent parameter, here the time before the coating is removed, is plotted on

the horizontal axis of a semi logarithmic diagram.

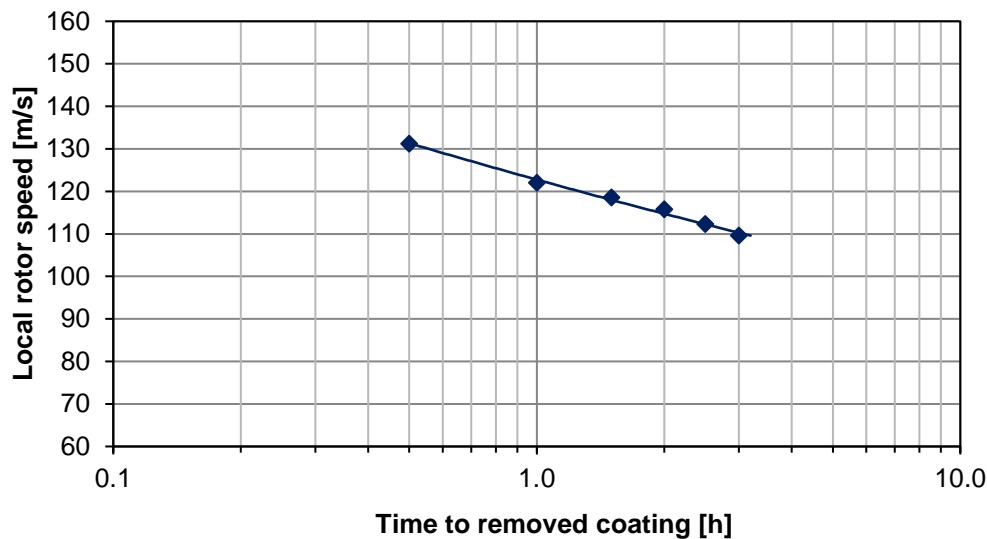

**Figure 3: Time to removed coating as a function of the local rotor speed.**

### 3.2 Generalizing empirical values

10  Exactly how the droplet sizes and velocities influence the damage is unknown. It obviously depends on several factors

including material configuration: properties of coating, putty and laminate, layer thicknesses, interfaces and various failure

modes. We now take the hypothesis that the kinetic energy of each impact characterizes the magnitude of the impact, and that the number of impacts per cm$^2$ corresponds to the number of cycles in a fatigue test.

For an object travelling through a rain field with assumed spherical droplets with uniform diameter and constant falling velocity, the impact frequency can be calculated analytically (Gohardani, O., 2011; DNVGL, 2018).

the relative volume of water in the rain field, *V,* is given by

$$V = \frac{I_r}{v_r} \tag{6}$$

where $I_r$ is rain intensity and $v_r$ is falling velocity.

The number of droplets per volume, *N,* can be expressed as

$$N = 6\frac{I_r}{v_r \pi D^3} \tag{7}$$

where *D* is droplet diameter.

An object travelling through a rain field is hit by droplets in a stochastic manner across its surface. Assuming that the droplets are distributed evenly in space, and their velocity is negligible compared to the speed of the object, the impact frequency or number of impacts per unit projected area per time, *F*, can be expressed as

$$F = N \cdot v_t \ [\text{s}^{-1}\text{m}^{-2}] \tag{8}$$

The kinetic energy, $E_k$, of each impact is

$$E_k = \frac{1}{12}\rho\pi D^3 v_t^2 \tag{9}$$

Now, the test data from table 1 can be presented as the number of impacts per unit area, $N_{EI}$, as a function of the kinetic energy of each impact before the coating is removed, see Fig. 4. Such a representation of data, often referred to as a Wöhler curve or SN curve, is known from fatigue testing of materials (Miner, 1945; Ronold and Echtermeyer, 1996), where the number of load

cycles causing failure is shown as a function of the magnitude of each load cycle (for example stress range). Fatigue data are often fitted to a power function as shown in Fig. 4.

$$N_{Ei} = F \cdot t = c \cdot (\frac{E_k}{E_0})^{-m} \tag{10}$$

$E_0 = 1$J. A power function fit of the test data in table 1 to Eq. (10) gives c=18 and m=4.63

We now take the additional hypothesis, that the incremental damage is a function of the impact energy only, and that Eq. (10)

holds for different droplet diameters.

Combining Eq. (9) and Eq. (10), the fatigue life can be expressed as a function of the impact velocity and the droplet diameter

$$N_{Ei} = c * \left(\frac{1}{12E_0}\rho\pi D^3 v_t^2\right)^{-m} \tag{11}$$

Now, applying Eq. (11), Wöhler curves can be drawn for different droplet diameters as shown in Fig. 5.

Back calculating from impacts per area to millimeters of accumulated rain, one gets the Wöhler curves for accumulated rain to remove the coating as a function of the rotor velocity for different droplet diameters as shown in Fig. 6. Given the assumptions and extrapolation it is obvious that droplet size is important and not just the amount of rain.

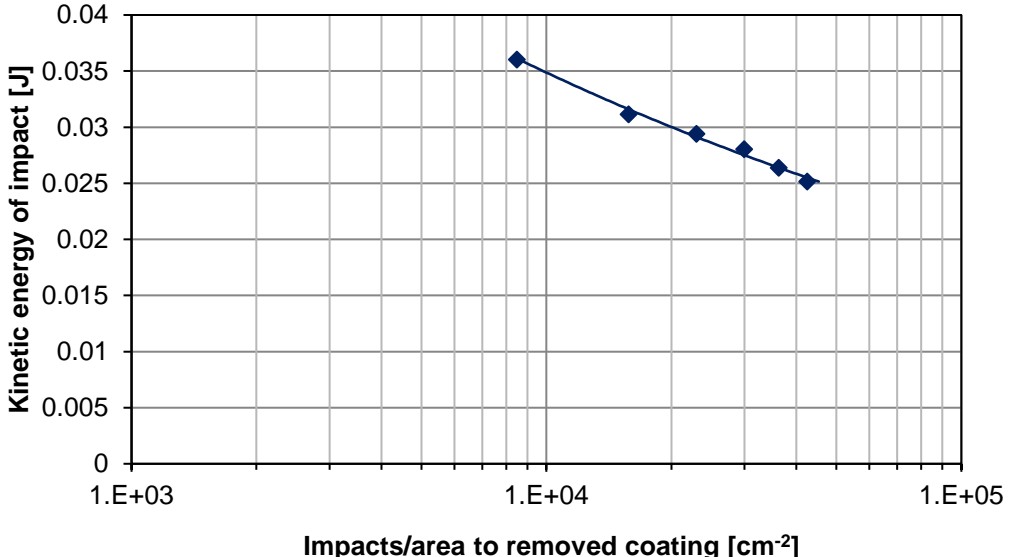

5 **Figure 4: Rain erosion test data plotted as a Wöhler curve: impacts per unit area to failure as function of the kinetic energy for each impact**

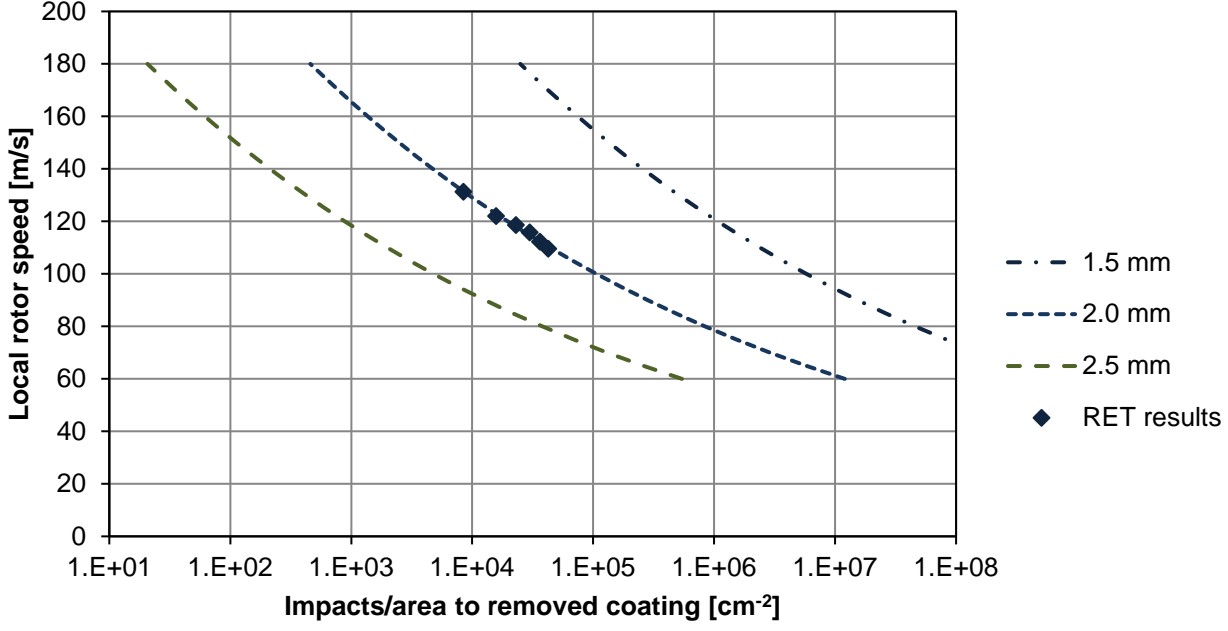

**Figure 5: Wöhler curves for droplet diameters of 1.5, 2.0 and 2.5 mm**

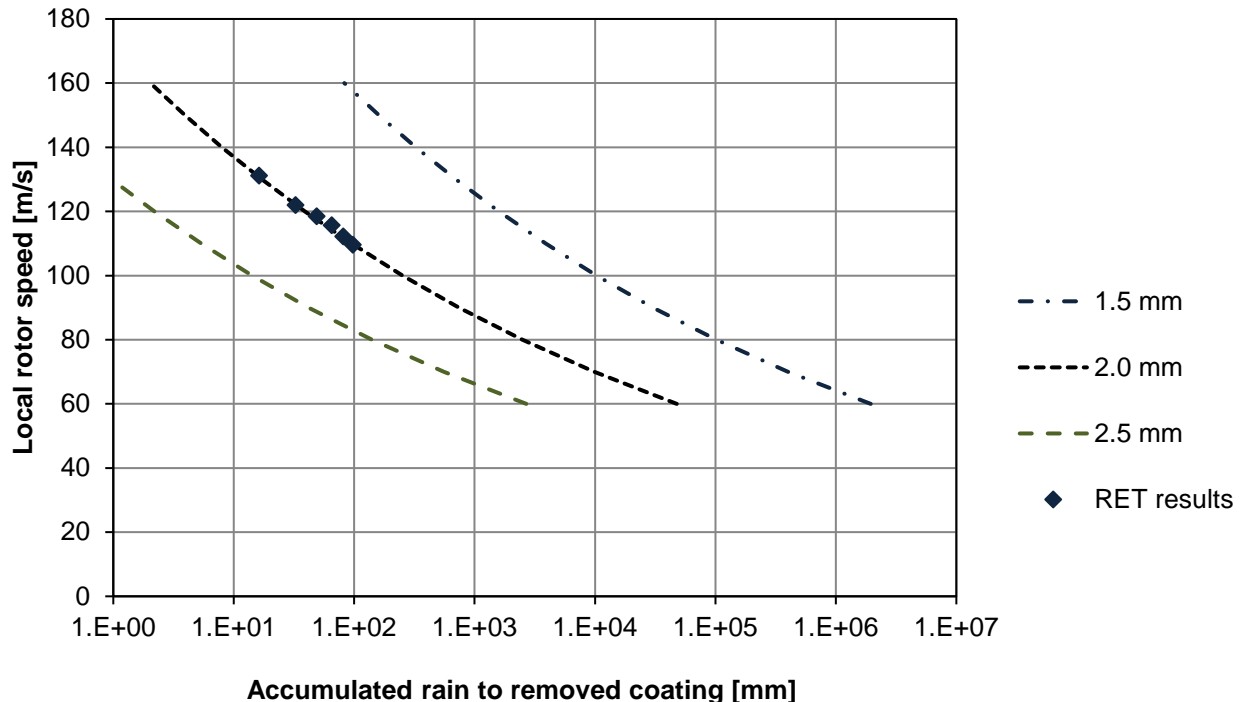

**Figure 6. Expected accumulated rain to remove coating as function of rotor speed for droplet diameters of 1.5, 2.0 and 2.5 mm**

### 3.3 Block loading and cumulative damage laws

Each point on the Wöhler curve in Fig. 3 corresponds to a test run at constant conditions (rain intensity, droplet size, local rotor speed). However, most structures designed for cyclic loads are subject to varying load intensities in service. For instance a wind turbine blade will see a spectrum of wind speeds, gusts and turbulence over its lifetime. Likewise a leading edge will be impacted with rain of varying droplet sizes and intensities and changing impact velocities. To account for variable conditions fatigue loading, different rules have been proposed for accumulation of damage in composites (Brøndsted et al., 1997). The most popular and easy to use, though not always correct, is the linear Palmgren-Miner rule. Accumulated damage (*M*) is given by

$$M = \sum_{i=1}^{j} \frac{n_i}{N_i} \tag{12}$$

Here $i$ is load level number, $n_i$ is number of cycles at level $i$, $N_i$ is cycles to failure at level $i$ in a test and $j$ is number of load levels. The expected fatigue life of a cyclic loaded material is reached when $M \geq 1$.

Given a load time history and Wöhler curves for different loading conditions, it is possible to use Miner's rule to determine the accumulated damage or fatigue life of a structure or material. The Palmgren-Miner rule has been used to predict the rain droplet impact fatigue life of a leading edge (Slot et al., 2015; Amirzadeh et al., 2017). Here it will be applied later to predict the fatigue life of a leading edge based on the presented RET data and rain statistics.

## 4 Precipitation

Estimation of the potential erosion caused by rain at specific wind farm sites has to be based on information on precipitation, wind speed and turbine characteristics such as tip speed. Wind speeds at wind farm sites are usually known from wind resource assessment during the planning phase and on site wind observations during the operational phase. In contrast, precipitation is not standard observation, neither for planning, nor for operation of wind farms.

Early studies on raindrop size distribution (Best, 1950; Mason and Andrews, 1960) showed rain intensity and raindrop size to relate to each other for a wide range of climate conditions. Fig. 7 shows the probability density function as a function of raindrop diameter for 6 rain intensities ranging from 0.1 to 20 mm/hr based on Best (1950) (Kubilay et al., 2013).

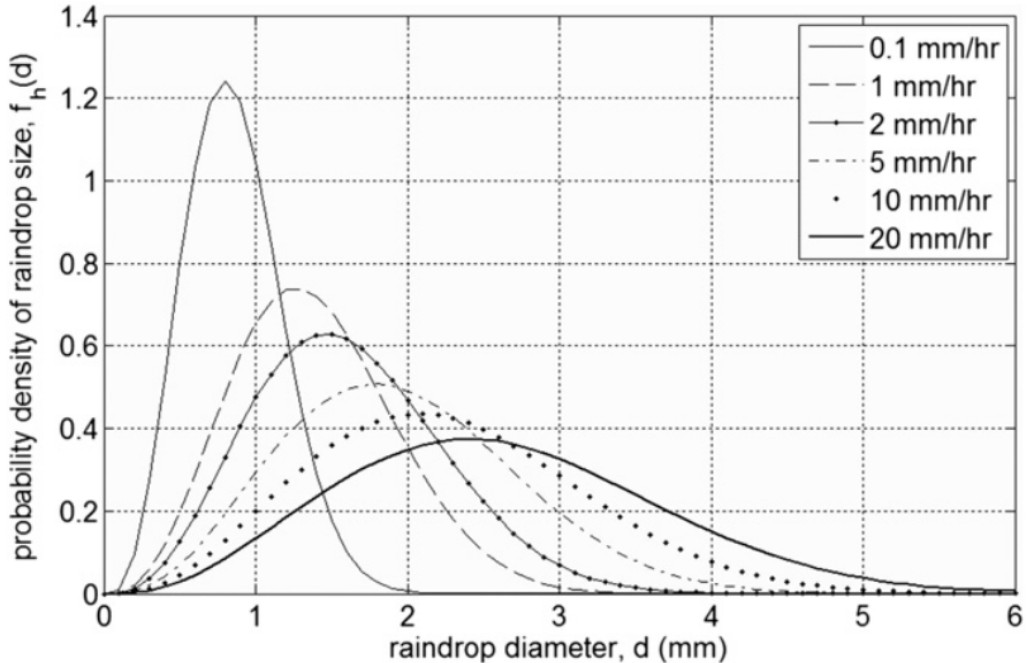

Figure 7: Raindrop size distribution through a horizontal plane with the rain fall intensity as a parameter (from Kubilay et al., 2013, based on Best, 1950).

Rain intensity (or rate) is typically measured as mm/h. Rain intensity varies a lot with time. The shorter the interval of measurement, the more detailed is the picture of variation. .

Based on disdrometer observations in New Jersey, USA, Smith et al. (2009) showed that raindrop size distribution and rainfall intensity in heavy convective rain can be described from a Gamma distribution. For a convective rain event, the rain intensity at 1 minute intervals can be more than 10 times higher than the intensity measured at 60 minutes intervals, (Smith et al., 2009). Convective rain is a type of precipitation, which is generally more intense, and of shorter duration, than rain from larger weather systems.

Similar results are found in tropical rainfall during the monsoon season in Malaysia (Hong et al., 2015). Precipitation measured by disdrometers at locations across the globe from Australia and Asia to Europe and America confirm the relationship between rain intensity and raindrop size distribution (Bringi et al., 2003). Interestingly, Bringi et al. (2013) distinguish between convective 'maritime' and convective 'continental' raindrop size distributions with the first being characterized by a lower

concentration of larger-sized drops as compared to the latter. The generalization on average rainfall rate and percentage of time of exceedance for different rain climates is shown in Fig. 8 (Jones and Sims 1978). Fig. 7 and Fig. 8 are used for input to droplet sizes and rain intensities for the simplified rain climate statistics used in section 5.1.2, table 4 to table 8.

Precipitation varies much across the globe. Mean annual precipitation and the monthly mean precipitation during the driest and wettest months are used in the Köppen-Geiger climate classification (Peel et al., 2007). The climatological standard normal

covers 30 years according to the World Meteorological Organization. The mean annual precipitation normal is based on local network station records on land and varies much spatially. Table 2 lists data from Scandinavia, the UK, Europe and the World. The wettest place on Earth is said to be in India with 11.871 mm per year (Source: World Atlas).

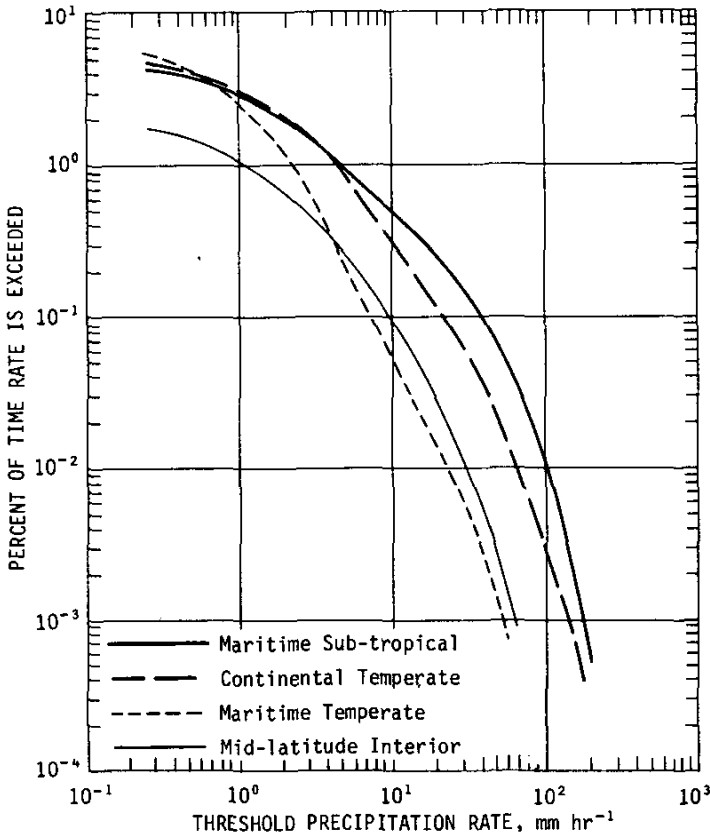

**Figure 8: Average rainfall rate-frequency relationships for four rain climates (Jones and Sims, 1978) ©American Meteorological**
**Society. Used with permission.**

**Table 2: Mean annual precipitation ranges over land in selected countries, Europe and the World.**

| Country | Range in mm | Period | Source |
|---|---|---|---|
| **Denmark** | <500 to >900 | 1961-1990 | Frich et al., 1997 |
| **Finland** | 400 to >800 | 1971-2000 | FMI |
| **Norway** | <300 to >4.000 | 1961-1990 | Met.no |
| **Sweden** | 400 to > 2.000 | 1961-1990 | SMHI |
| **UK** | <400 to > 3.000 | 1981-2010 | Met Office |
| **Europe** | <300 to > 4.000 | n.a. | http://i.imgur.com/kEJhdOK.jpg; Panagos et al., 2015 (their Fig.1) |
| **World** | <50 to > 11.000 | n.a. | WorldClim; Climate-Charts.com;GPCC |

Precipitation over the ocean is mapped mainly by Earth observing satellites and to lesser degree based on sparse observations from ships and weather stations on small islands. The annual precipitation during the years from 1998 to 2011 observed by the

Tropical Rainfall Microwave Mission (TRMM) between latitudes 40°N and 40°S is shown in Fig. 9. The spatial resolution is 0.25° by 0.25°. Annual rainfall up to 7300 mm is noted in some tropical regions over ocean. Over land the TRMM map shows dry and wet regions corresponding to precipitation maps based on weather stations.

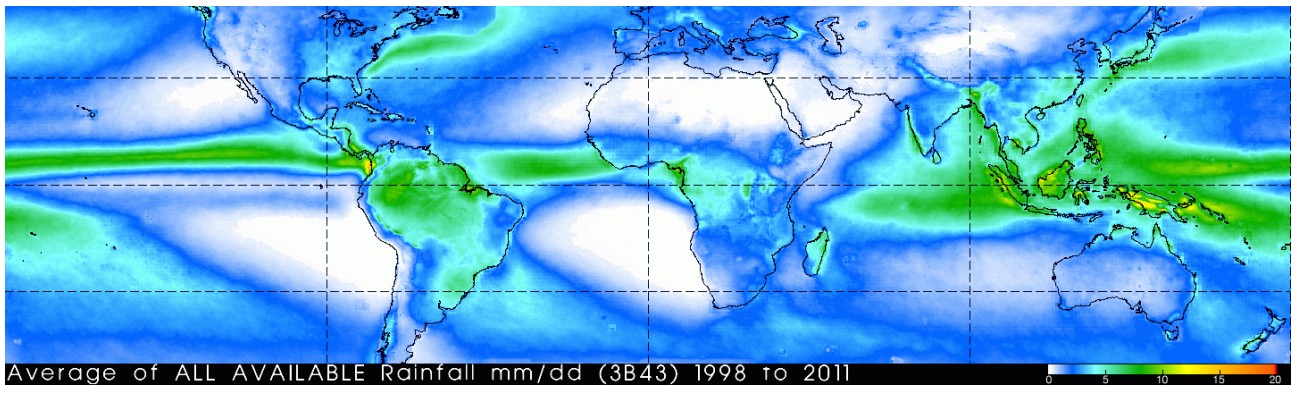

**Figure 9. Average rainfall measured by TRMM from 1998 to 2011. Source: NASA.**

The global precipitation map covering the years from 1988 to 2004 is shown in Fig. 10. This map is based on the Special Sensor Microwave Imager, GOES precipitation index, outgoing longwave precipitation index, rain gauges, and sounders on NOAA satellites (Source: GPCP). The spatial resolution is 2.5° by 2.5°. The map shows, that annual precipitation is above 3300 mm per year over the ocean in some tropical regions. It may be noted that this spatial resolution does not resolve details. The maps for Scandinavia, the UK, Europe and the World listed in table 2 do not cover the sea. TRMM only covers between

40°N and 40°S. Thus a map of the 30 year mean annual precipitation in the Northern European Seas, where the majority of offshore wind farms are located, is not available (to the knowledge of the authors).

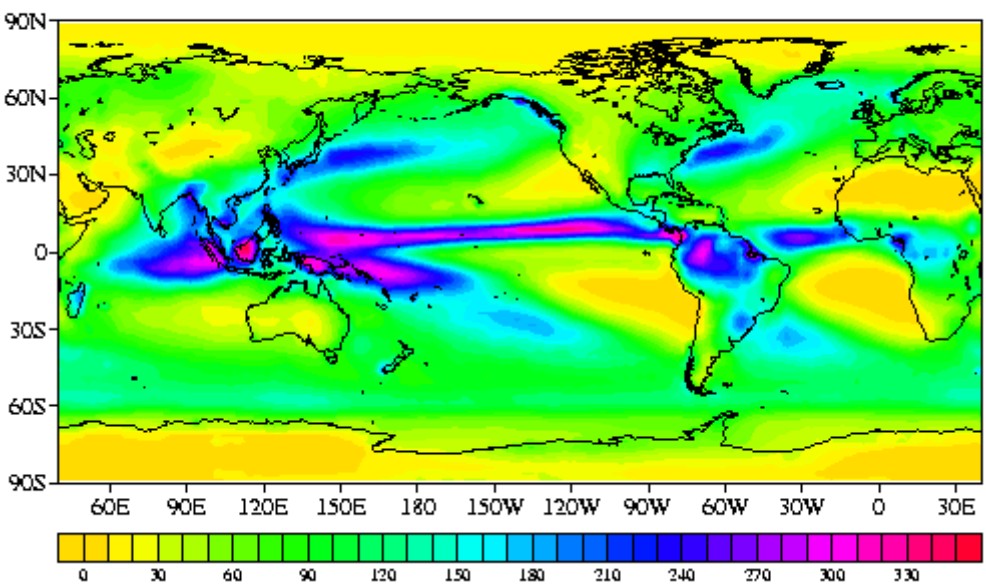

**Figure 10: Average rainfall measured by several satellites, sounders and rain gauges combined for the years 1988 to 2004. Source: GPCP.**

The objective to estimate the potential erosion caused by rain at specific wind farm sites is obviously more challenging at sea than at land due to the limited available precipitation data. Over land the rainfall erosivity for soil degradation has been assessed from weather station data (Panagos et al., 2015; Panagos et al., 2017). It is based on the Revised Universal Soil Loss Equation (RUSLE) method (Naipal et al., 2015). Rainfall erosivity is modelled as a function of the kinetic energy of rain, the maximum intensity of rainfall, the cumulative rainfall, the soil properties and the slopes of terrain. The map on rainfall erosivity in Europe at 500 m spatial resolution assessed by European Soil Data Centre (ESDAC) is shown in Fig. 11 (Panagos et al., 2015). A comprehensive precipitation database is established (Source: ESCAC, Panagos et al., 2015; Panagos et al., 2017). This database would be valuable for the production of a rain erosion map for wind turbines where precipitation, wind speed and turbine characteristics such as tip speed would be input.

For offshore wind farms the leading edge life is significantly shorter, than what is observed on land due to longer blades and higher tip speeds offshore (Cortés et al. 2017), and likely also due to offshore rain and wind conditions, ocean salinity and marine air composition. The wind turbine blades offshore need inspection and repair during life time. The access to offshore wind farms is dependent upon suitable weather conditions and the cost of keeping staff and machinery waiting for the right weather window can be significant (Poulsen et al., 2017). During repair with leading edge protection on offshore blades the weather window require benign wind and wave plus additionally air temperatures above 15°C, relative humidity <60% and no warning for thunderstorm and lightning. In the Northern European Seas this limits repair campaigns to the summer period. It

may be valuable to assess the likelihood of suitable weather windows, in addition to the wind resource and the potential rain erosion for improved overall assessment of life time cost for offshore wind farms.

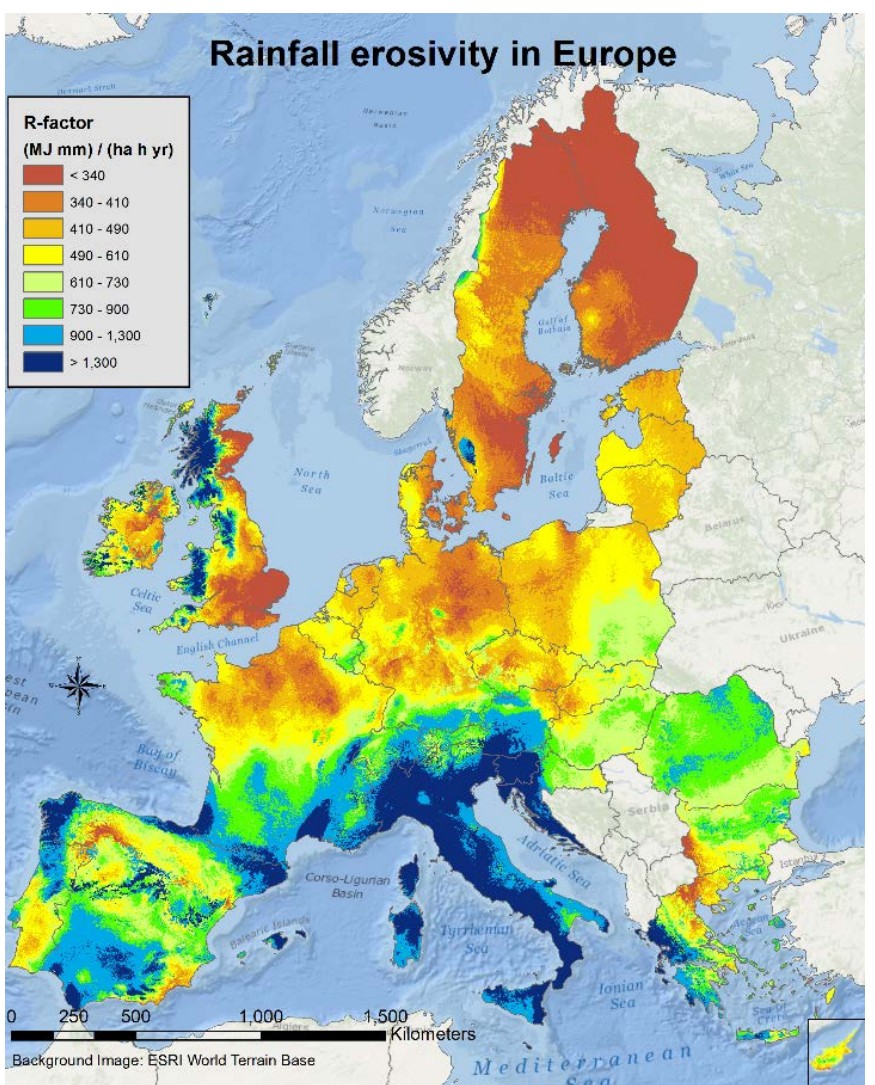

**Figure 11: Rainfall erosivity in Europe at 1-km gric cell resolution. Source: Panagos et al., 2015. Creative Commons Attribution-NonCommercial-No Derivatives License (CC BY NC ND), https://doi.org/10.1016/j.scitotenv.2015.01.008**

## 5 Turbine control for reducing tip speed

Leading edge erosion causes an increase in surface roughness of the blade and thereby an increase in the air flow boundary layer thickness over the airfoils on the blade when it is operating. The increased boundary layer thickness causes increased drag coefficient and decreased lift coefficient, and thus reduces the aerodynamic performance, particularly at higher angles of attack (AOA) (Sareen et al., 2014). The consequence is severe losses in energy production. To investigate the influence of the

erosion on the aerodynamic performance and on the annual energy production (AEP), aerodynamic rotor computations were carried out for a Vestas V52 wind turbine with a modified control system. First, the method is described, and then the results are shown.

## 5.1 Methods

### 5.1.1 The wind turbine

The investigation was carried out as simulations on the Vestas V52 850 kW pitch regulated variable speed wind turbine that was erected at the DTU Risø Campus during the summer of 2015, table 3. This wind turbine was chosen because data was available. However, parts of the input were modified, e.g. the rotational speed to make it consistent with the higher tip speeds that modern wind turbines are designed with. The size of the wind turbine is somewhat smaller than the majority of wind turbines installed during the last decade, but the relative losses in annual energy production are considered similar. The simulations were carried out assuming steady state, no yaw and no aeroelastic response. These assumptions simplified the conditions significantly, but were made to investigate the main response. A further description of the simulations is found in section 5.1.3 to section 5.1.5.

**Table 3: Data for the Vestas V52 wind turbine.**

| Technical data for the Vestas V52-850kW | |
| --- | --- |
| Power regulation | Variable speed / variable pitch |
| Number of blades | 3 |
| Rotor diameter | 52 [m] |
| Hub height | 44 [m] |
| Maximum rotor speed | 33 rpm (assumed) |

### 5.1.2 Control of the wind turbine

The wind turbine control is of the pitch regulated variable speed type. The maximum tip speed is assumed to be 90m/s, and is thereby greater than the tip speed of an original Vestas V52 wind turbine. Compared to common control strategies, this wind turbine is assumed to have a precipitation sensor. The sensor is capable of measuring and giving input to the turbine controller regarding the present rain intensity and/or the droplet size. As the tip speed is a governing factor in the erosion rate, the erosion control strategy consists in reducing the maximum tip speed, when the rain intensity and droplet size is exceed threshold values. This is done by reducing the maximum rotational speed of the wind turbine. Because a wind turbine is designed e.g. for maximum torque in the drive train, the reduction of the rotational speed implies that the maximum power is reduced. The wind turbine with the original control can produce a maximum mechanical power described by $P_0 = Q_0 \cdot \omega_0$ , where $P_0$ is the original rated mechanical power, $Q_0$ is the original rated main shaft torque and $\omega_0$ is the original maximum rotational speed. When reducing the rotational speed from $\omega_0$ to $\omega_1$, the maximum torque limit must not be exceeded. Then

the maximum power is reduced as well to $P_1$, $P_1=Q_0 \cdot \omega_1$. An example from the computations is shown in Fig. 12. It is seen that the maximum torque is maintained despite of the difference in maximum rotational speed.

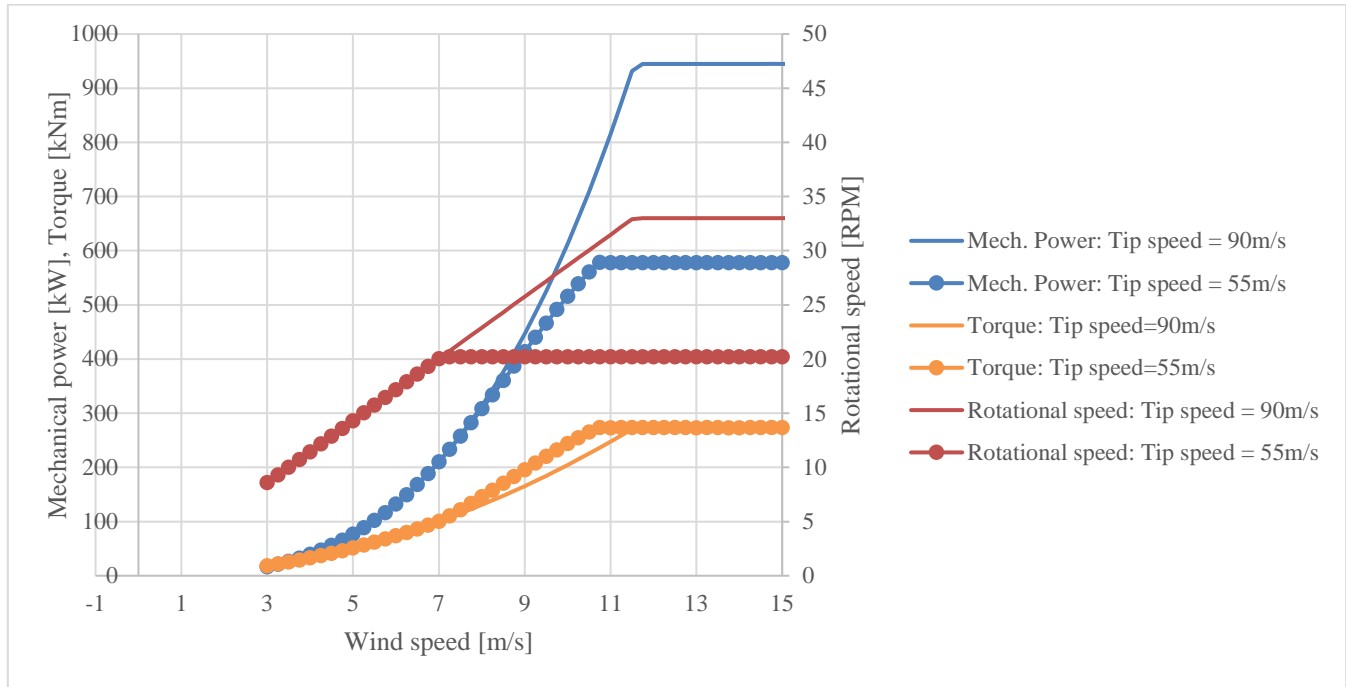

**Figure 12: Mechanical power, rotational speed and main shaft torque as functions of wind speed for control of the rotor with maximum tip speeds of 90m/s and 55m/s respectively.**

Five different erosion control strategies (ECS) are investigated. An ECS is a set of one or more precipitation intensity threshold values and the corresponding maximum allowed tip speeds, (max tip speed@rain intensity threshold). The expected lifetime for the blade leading edge for each ECS is calculated using the droplet size dependent Wöhler curves, Eq. 11, Palmgren Miners rule, Eq. 12, and the assumed rain data, which are deduced from Fig. 7 and Fig. 8 and shown in the columns 1, 2 and 3 of table 4 to table 8. Additionally a reference case is included, where it is assumed, that no erosion occurs.

- ECS 1: No tip speed reduction; expected life time of 1.6 years
- ECS 2: (70 m/s@20 mm/hr); (80 m/s@10 mm/hr); expected life time of 10 years
- ECS 3: (60 m/s@20 mm/hr); (70 m/s@10 mm/hr); expected life time of 24 years
- ECS 4: (60 m/s@20 mm/hr); (70 m/s@10 mm/hr); (70 m/s@5 mm/hr); expected life time of of 54 years
- ECS 5: (55 m/s@20 mm/hr); (65 m/s@10 mm/hr); (70 m/s@5 mm/hr); expected life time of 107 years
- Reference strategy 6 with no tip speed reduction and expected life time of infinite many years

The results of the five first control strategies are shown in tables 4 to 8. For the reference strategy 6, it is assumed that no erosion will occur. The rain intensity frequencies are based on precipitation data for maritime temperate climate from Fig. 8. The fixed droplet sizes for each rain intensity are assumed based on Fig. 7. The expected life times are calculated applying equations 11 and 12 and extensive extrapolation as described in section 3 of the RET data shown in table 1. The control strategies are based on an assumed behavior that there is correspondence between the surface roughness height and the

aerodynamic performance. Thus, there are elements in this analysis that are not documented, but are based on qualified assumptions. However, the numbers are believed to be sufficiently realistic to demonstrate the potential of erosion control.

The first row in table 4 is explained here: the probability of rain intensity of 20 mm/hour with 2.5 mm droplets is around 0.02% or 1.8 hour per year. At this rain intensity the total expected lifetime before failure of the leading edge (LE) at 90 m/s is around 3.5 hours. The fraction of damage per year relative to LE failure at this level is 51%. Summing up the fractions of damage per year at the five different rain intensities gives 0.64 or 64%. The calculated expected blade LE life at these conditions is therefore 1/0.64=1.6 year.

The results in table 5 show the effect of reducing the tip speed to 70 m/s and 80 m/s, respectively, at the two heaviest rain intensities. Because of the reduction of tip speed the blade life is extended to 10 years. In tables 6 to 8 the results are shown where further increase of lifetime is obtained.

**Table 4: Calculation of the expected life time of the blade leading edge with no reduction of the tip speed. Control strategy 1.**

| Rain intensity [mm/hr] | Droplet size [mm] | Percent of time [%] | Hours per year [hrs/year] | Blade tip speed [m/s] | Time to LE failure [hrs] | Fraction of life spent per year [%] | |
|---|---|---|---|---|---|---|---|
| 20 | 2.5 | 0.02 | 1.8 | 90 | 3.5 | 51 | |
| 10 | 2.0 | 0.1 | 8.8 | 90 | 79 | 11 | |
| 5 | 1.5 | 1 | 88 | 90 | $3.6 \cdot 10^3$ | 2.4 | |
| 2 | 1.0 | 3 | 263 | 90 | $7.5 \cdot 10^5$ | 0.0 | |
| 1 | 0.5 | 5 | 438 | 90 | $2.8 \cdot 10^9$ | 0.0 | |
| | | | | Sum of fractions [%]: | | 64 | |
| | | | | Expected life [years]: | | | 1.6 |

**Table 5: Calculation of the expected life time of the blade leading edge with reduction of the tip speed to 70m/s and 80m/s, respectively: Control strategy 2**

| Rain intensity [mm/hr] | Droplet size [mm] | Percent of time [%] | Hours per year [hrs/year] | Blade tip speed [m/s] | Time to LE failure [hrs] | Fraction of life spent per year [%] | |
|---|---|---|---|---|---|---|---|
| 20 | 2.5 | 0.02 | 1.8 | 70 | 46 | 3.8 | |
| 10 | 2.0 | 0.1 | 8.8 | 80 | 263 | 3.3 | |
| 5 | 1.5 | 1 | 88 | 90 | $3.6 \cdot 10^3$ | 2.4 | |
| 2 | 1.0 | 3 | 263 | 90 | $7.5 \cdot 10^5$ | 0.0 | |
| 1 | 0.5 | 5 | 438 | 90 | $2.8 \cdot 10^9$ | 0.0 | |
| | | | | Sum of fractions [%]: | | 9.6 | |
| | | | | Expected life [years]: | | | 10.4 |

**Table 6: Calculation of the expected life time of the blade leading edge with reduction of the tip speed to 60m/s and 70m/s, respectively: Control strategy 3**

| Rain intensity [mm/hr] | Droplet size [mm] | Percent of time [%] | Hours per year [hrs/year] | Blade tip speed [m/s] | Time to LE failure [hrs] | Fraction of life spent per year [%] |
|---|---|---|---|---|---|---|
| 20 | 2.5 | 0.02 | 1.8 | 60 | 222 | 0.8 |
| 10 | 2.0 | 0.1 | 8.8 | 70 | $1.0 \cdot 10^3$ | 0.8 |
| 5 | 1.5 | 1 | 88 | 90 | $3.6 \cdot 10^3$ | 2.4 |
| 2 | 1.0 | 3 | 263 | 90 | $7.5 \cdot 10^5$ | 0.0 |
| 1 | 0.5 | 5 | 438 | 90 | $2.8 \cdot 10^9$ | 0.0 |
| | | | | Sum of fractions [%]: | | 4.1 |
| | | | | Expected life [years]: | | 24 |

**Table 7: Calculation of the expected life time of the blade leading edge with reduction of the tip speed to 60m/s, 70m/s and 70m/s, respectively: Control strategy 4**

| Rain intensity [mm/hr] | Droplet size [mm] | Percent of time [%] | Hours per year [hrs/year] | Blade tip speed [m/s] | Time to LE failure [hrs] | Fraction of life spent per year [%] |
|---|---|---|---|---|---|---|
| 20 | 2.5 | 0.02 | 1.8 | 60 | 222 | 0.8 |
| 10 | 2.0 | 0.1 | 8.8 | 70 | $1.0 \cdot 10^3$ | 0.8 |
| 5 | 1.5 | 1 | 88 | 70 | $4.8 \cdot 10^4$ | 0.2 |
| 2 | 1.0 | 3 | 263 | 90 | $7.5 \cdot 10^5$ | 0.0 |
| 1 | 0.5 | 5 | 438 | 90 | $2.8 \cdot 10^9$ | 0.0 |
| | | | | Sum of fractions [%]: | | 1.9 |
| | | | | Expected life [years]: | | 54 |

**Table 8: Calculation of the expected life time of the blade leading edge with reduction of the tip speed to 55m/s, 65m/s and 70m/s, respectively: Control strategy 5**

| Rain intensity [mm/hr] | Droplet size [mm] | Percent of time [%] | Hours per year [hrs/year] | Blade tip speed [m/s] | Time to LE failure [hrs] | Fraction of life spent per year [%] |
|---|---|---|---|---|---|---|
| 20 | 2.5 | 0.02 | 1.8 | 55 | 541 | 0.3 |
| 10 | 2.0 | 0.1 | 8.8 | 65 | $2.2 \cdot 10^3$ | 0.4 |
| 5 | 1.5 | 1 | 88 | 70 | $4.8 \cdot 10^4$ | 0.2 |
| 2 | 1.0 | 3 | 263 | 90 | $7.5 \cdot 10^5$ | 0.0 |
| 1 | 0.5 | 5 | 438 | 90 | $2.8 \cdot 10^9$ | 0.0 |
| | | | | Sum of fractions [%]: | | 0.9 |
| | | | | Expected life [years]: | | 107 |

Not to overload the drivetrain it was ensured not to exceed the maximum rated shaft torque. Thus, when operating at different maximum tip speeds the wind turbine had to operate at different rated power:

- 90m/s: 850kW
- 80m/s: 760kW
- 70m/s: 660kW
- 65m/s: 615kW
- 60m/s: 570kW
- 55m/s: 520kW

Even though the wind turbine experiences heavy rain and have to reduce the tip speed, the wind turbine will produce some power; thus only part of the potential power is lost. On the other hand, by using the erosion safe mode the repair and loss in production due to leading edge erosion will be reduced.

### 5.1.3 Determination of the loss in annual energy production

The prediction of the rotor performance was based on a design tool, HAWTOPT, for multi point wind turbine design.. The tool is basically a Blade Element Momentum (BEM) code with the ability also to compute e.g. energy production, and with the further ability also to optimize the operational data (i.e. pitch and RPM), using numerical optimization. HAWTOPT was used to calculate the aerodynamic performance of the Vestas V52 rotor given different sets of airfoil characteristics corresponding to different degrees of erosion. For further information about HAWTOPT, see Fuglsang et al. (2001). HAWTOPT calculated the annual energy production based on the power curve that is a result of the BEM computation and a Weibull distribution, where the mean wind speed is varying, so that A=7m/s, 8m/s and 9m/s, and where the shape is constant, C=2. The airfoil characteristics for the blades in terms of lift coefficients, drag coefficients and moment coefficients as a function of angles-of-attack were predicted as described in the section 5.1.4. It should be emphasized that HAWTOPT only takes into account the steady state aerodynamics. Even though more extensive investigations could have been carried out with an unsteady aeroelastic code, so that the load response (ultimate and fatigue) was evaluated as well, it was the main aerodynamic mechanisms that were investigated. Thus, aeroelastic tools (Øye, 1996; Bossanyi, 2004; Lindenburg et al., 2000; Jonkman et al., 2005; Larsen et al., 2005), with detailed simulation of the control system, e.g. (Bossanyi 2003) were not used and therefore the word "control" as described in this work is used as a broader term describing the tip speed and rated power. Thus, in this work, control algorithms are not included. But combined with the steady state aerodynamic computations, the operational data, rotational speed and pitch, are optimized to obtain maximum power coefficient at any maximum allowed rotational speed.

### 5.1.4 Method for derivation of aerodynamic airfoil data

Flow computations were carried out using *XFOIL* (version 6.1) developed by Drela (1989), because wind tunnel tests were not available for all airfoil sections on the blade. The computations were done for the angle-of-attack range from -20° to 20°, and the transition point from laminar to turbulent flow was modeled as free transition by the $e^n$ method and forced transition

setting $x/c_c$=0.1% at suction side and $x/c_c$ =10% at the pressure side. In the $e^n$ model the amplification factor value $n$=7 was used because this corresponds to a turbulence intensity of around 0.1%, which is common for high quality wind turbine blades and because this value has shown to predict the transition point position well compared to tunnel tests and atmospheric flow. Finally, the airfoil characteristics (except for the cylinder part) are 3D corrected according to Bak et al. (2006).

5 An example of a set of derived data is shown in Figure 13, where the airfoil characteristics for a relative thickness of $t/c_c$=15%, corresponding to the outer part of the blade, , are seen. To the left, plots of lift coefficient, $c_l$, as a function of drag coefficient, $c_d$, are seen. To the right lift coefficient, $c_l$, as a function of angle of attack is seen. The blue curves show the performance for perfectly clean (none-eroded) airfoils, whereas the red curves show the performance for airfoils with full leading edge roughness (LER) that corresponds to full blade life for the airfoil as stated in tables 4 to 8. An example of an airfoil performance

10 at 60% of full LER is shown with the green curves. The green curves are simple interpolations between the curves for perfectly clean airfoils and airfoils with full LER.

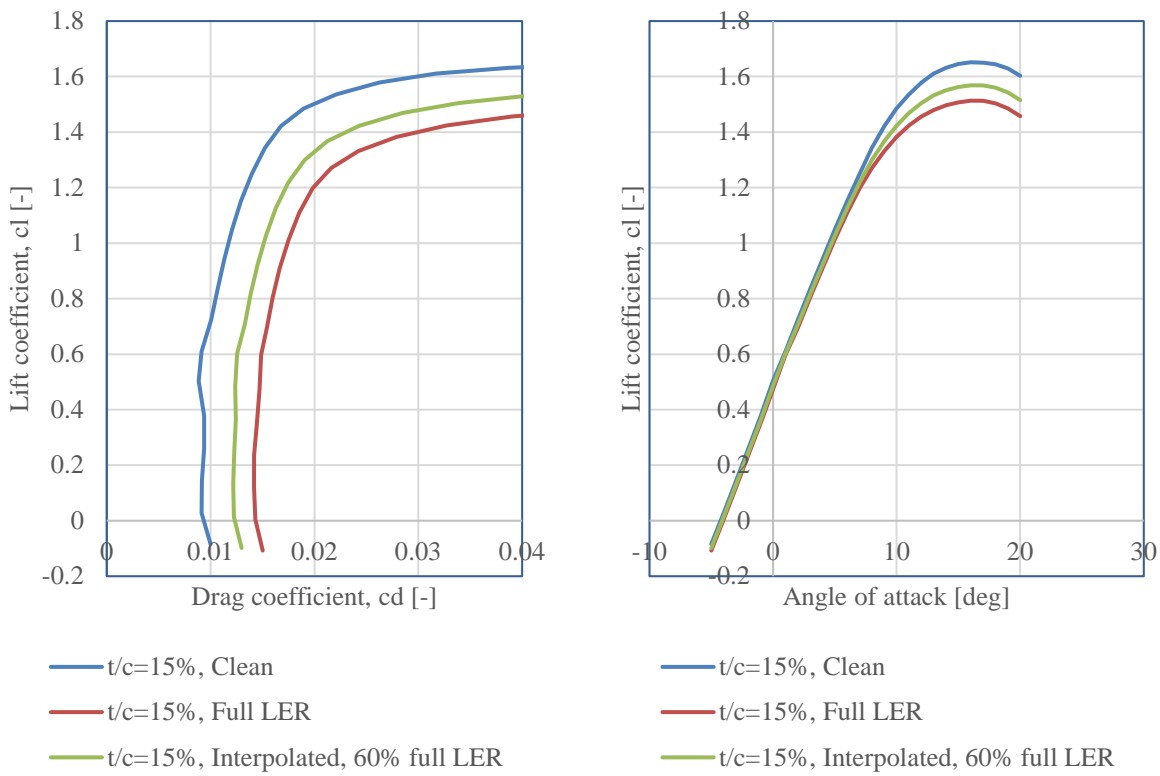

**Figure 13: Airfoil characteristics of the outer part of the blade: blue = clean blade , red = full leading edge roughness and green = 60% of full leading edge roughness. Left: Lift coefficient, cl, as a function of drag coefficient, cd. Right: Lift coefficient, cl, as a function of angle-of-attack, AOA.**

Power curves for different levels of LER are shown in Fig. 14, where the clean blade, the blade with full LER and some of the intermediate roughness levels are reflected. Thus, the intermediate roughness levels represent the corresponding LE lifetime, so e.g. 20% of full LER corresponds to 20% of the lifetime. This correspondence is not documented and is therefore a postulate that is however based on experience. In the analysis, roughness levels with steps of 10% difference are used.

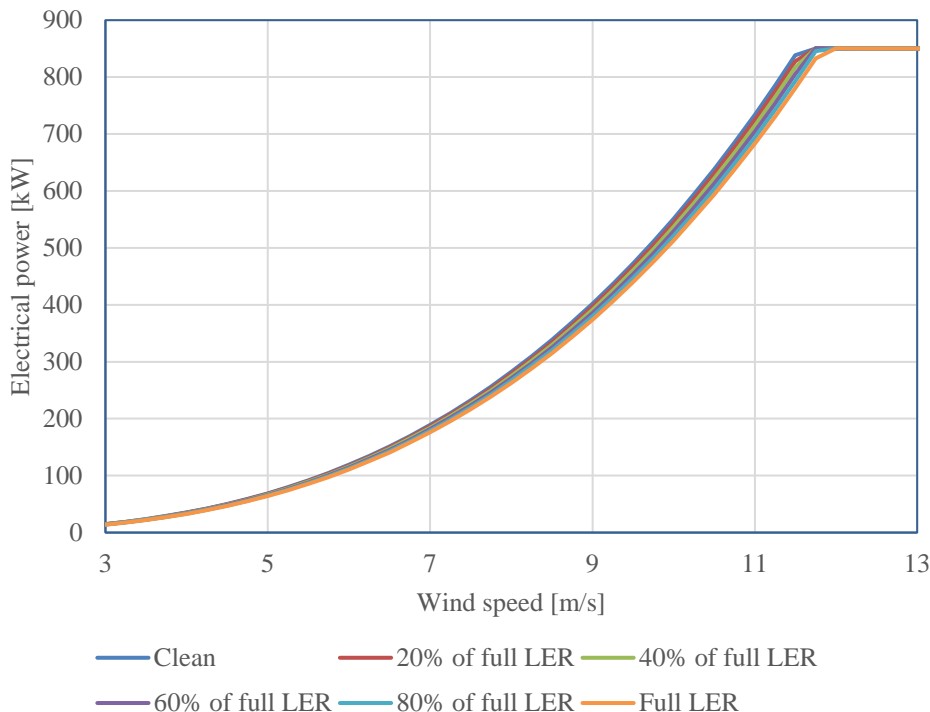

**Figure 14: Simulated power curves for the Vestas V52 for different leading edge roughness levels.**

Power curves for different maximum allowed tip speeds are shown in Fig. 15. The plot shows how the power curves change when the tip speed is reduced, not to overload the shaft torque. It shows that the power curves are almost identical from wind speeds of 3m/s to 9m/s. Thus, reduction of the rated power will influence the production for wind speeds greater than 9m/s.

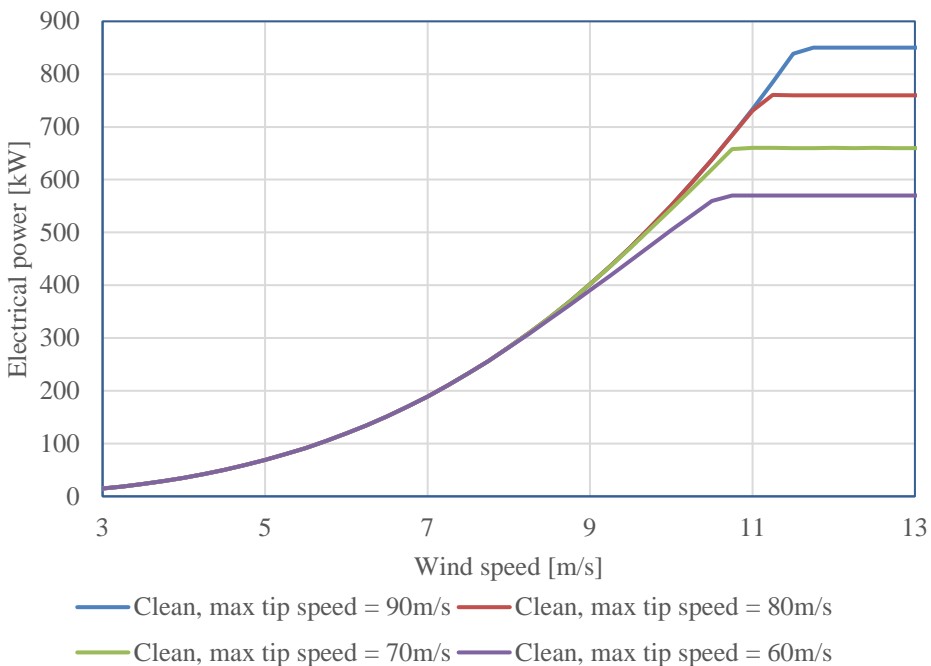

**Figure 15: Simulated power curves for the Vestas V52 for different maximum tip speeds.**

### 5.1.5 Cost of operation and maintenance

The selling price of energy, and the costs and down time of inspection and repair are assumed based on discussions with industrial partners. These values can vary a lot.

- Energy price:
  - 50 €/MWh
  - 250 €/MWh
- Inspection cost:
  - 500 €/rotor
  - 1500 €/rotor
- Repair cost
  - 10000 €/rotor
  - 20000 €/rotor

Apart from these costs, there is also a loss in production due to stand still of the rotor. The following stand still is assumed for the different control strategies, where a stand still of 1 day when inspected and a stand still of 2 days when repaired are assumed:

- Control strategy 1: 10 inspections and 9 repairs
- Control strategy 2: 10 inspections and 1 repairs
- Control strategy 3: 5 inspections and 0 repairs
- Control strategy 4: 5 inspections and 0 repairs
- Control strategy 5: 2 inspections and 0 repairs
- Control strategy 6: 2 inspections and 0 repairs

Based on these prices and costs, the cases in tables 4 to 8 are evaluated in the next section.

## 5.2 Results

Based on the assumed rain climate and the five erosion control strategies, table 4 to table 8, the aerodynamic modeling and the cost of energy, inspection and repair, the overall loss of income due to leading edge erosion and its mitigation is calculated for the different erosion control strategies. The energy production is computed by dividing the turbines energy production over the life time into 10 sections with different power curves. The power curve for the clean (non-eroded) rotor is valid for the first 10th of the lifetime. The power curve with 10% of full leading edge roughness is valid for the next 10th of the lifetime, the power curve with 20% of full leading edge roughness is valid for the next 10th of the lifetime and so on. When the lifetime is reached, the turbine will operate with the full leading edge roughness until the next whole year has past, and then it will be repaired. E.g. with a life time of 1.6 years the rotor will be repaired after 2 years, because blade repairs are mainly carried out during the summer, when temperature, humidity and wind are occasionally appropriate. After repair, it is assumed that the blades are completely clean and that they are as wear-resistant as new blades. Three sources for the loss of energy production are taken into account: Losses due to degradation in aerodynamic performance, losses due to standstill during inspection and repair and finally the losses due to the occasional reduction in maximum tip speed. It is assumed, that the duration of tip speed reduction is three times the duration of the heavy precipitation event, because the turbine cannot react instantaneously.

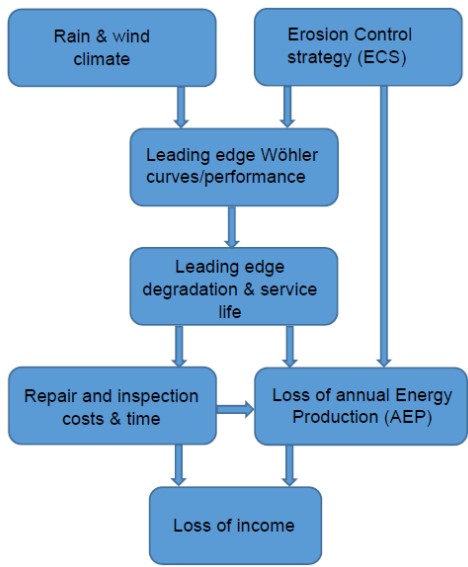

**Figure 16: Illustration of the framework for calculating the loss of income due to leading edge erosion.**

Fig. 16 illustrates in a simplified manner the framework developed here as a tool for selecting the erosion control strategy (ECS) for minimizing the loss of income due to leading edge erosion, The site specific rain and wind statistics, the operational data and ECS determine the erosive loads on the leading edge. Together with the rain erosion test Wöhler curves these loads are used as input to the cumulative damage rule (Palmgren & Miner) to compute the expected service life. The aerodynamic degradation, the intervals for inspection and repair and the loss of production due to occasional tip speed reduction are then

used to minimize the loss of income. In field operation, precipitation sensors give input to the control system of the turbines to apply the ECS and adapt to present weather conditions.

In Fig. 17 the annual energy production (AEP) including stand still due to inspection and repair is reflected. It is seen, that applying control strategy 1, where there is no reduction of the maximum tip speed, the loss of AEP is significant with up to 3.5% compared to the reference case with no erosion and no inspections and repairs. The loss of AEP is clearly dependent on the wind climate. For low wind speed sites with A=7m/s the loss is greater than for sites with higher wind speeds, A=9m/s.

In Fig. 18 to Figure 21 the loss of income due to lost AEP, inspection and repair are seen with the assumption of different costs of energy, inspection and repair. In the plots the loss of income is significantly simplified and is computed as

$$Income = TEP*EC – NOR*CR – NOI*CI, \tag{13}$$

where TEP is the total energy production [kWh], EC is the energy cost [€/kWh], NOR is the number rotor repairs during the life of the turbine, CR is the cost of the repair [€/rotor], NOI is the number of rotor inspections and CI is the cost of each inspection [€/rotor]. From the plots it is seen, that the loss of income can be significant. The income is very dependent on the energy price and the cost of repair, but a clear trend is that the erosion safe mode increases the income. Even the very advanced erosion safe mode, control strategy 5, with rather low tip speeds results in a significant improvement. As an example, control strategy 2 can be investigated. Here, the tip speed is reduced from 90m/s to 70m/s during 5.4 hours/year and from 90m/s to 80m/s during 26.4 hours/year due to heavy precipitation. In this case AEP is increased with around 1% and the income loss is decreased from 15.4% to 4.5% in the worst case and from 4.7% to 2.7% in the best case, depending on assumptions in cost and wind climate.

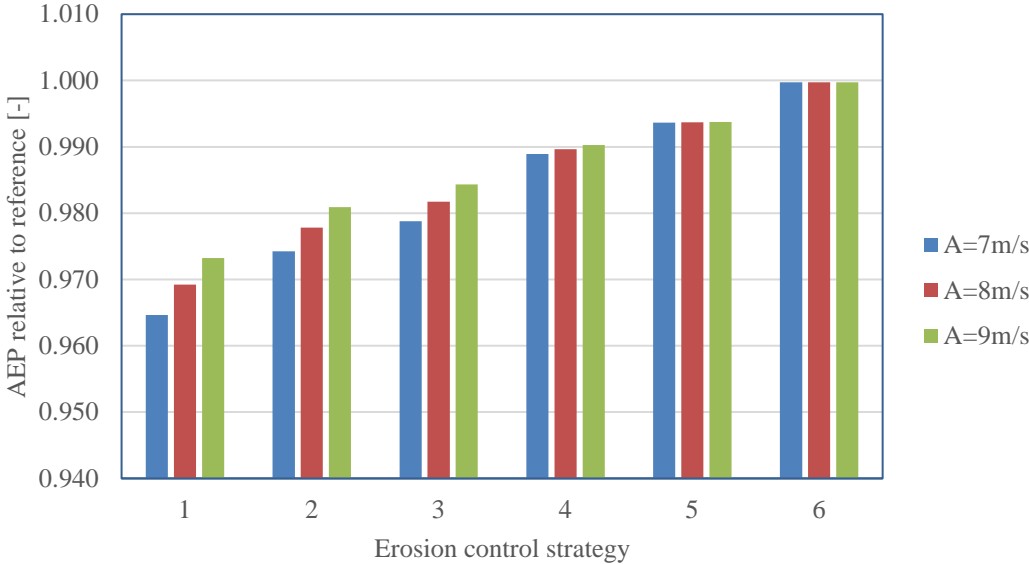

**Figure 17: AEP relative to AEP with no erosion.**

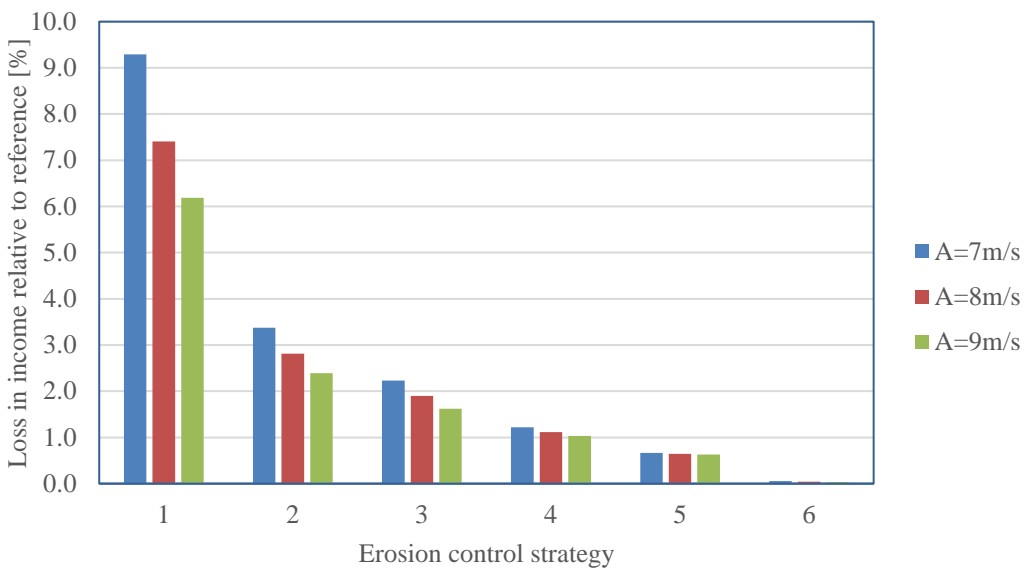

**Figure 18: Loss of income due to erosion, inspection and repair. Energy: 50€/MWh]. Repair: 10000€/rotor. Inspection: 500€/rotor.**

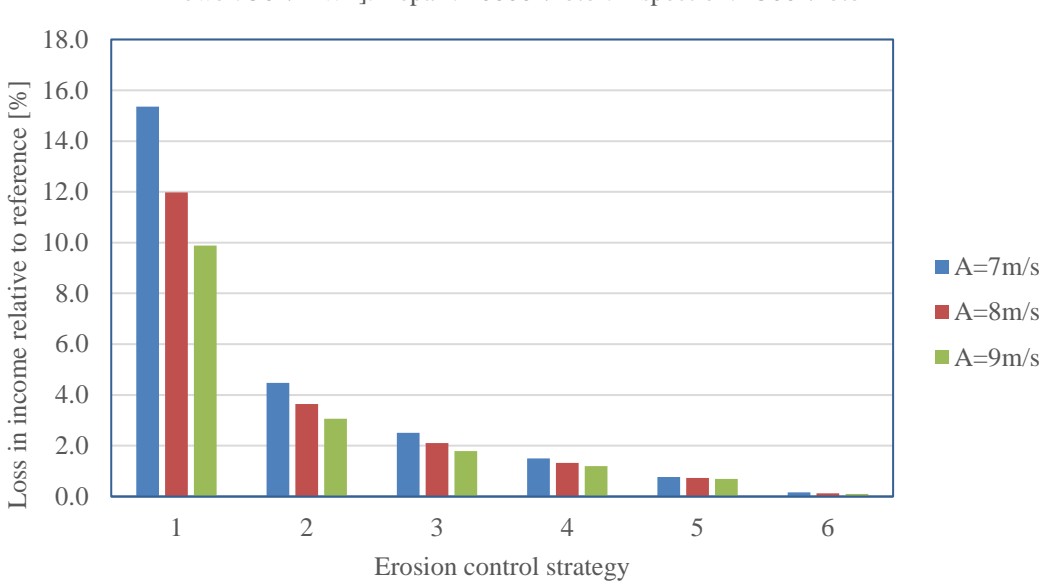

**Figure 19: Loss of income due to erosion, inspection and repair. Energy: 50€/MWh]. Repair: 20000€/rotor. Inspection: 1500€/rotor**

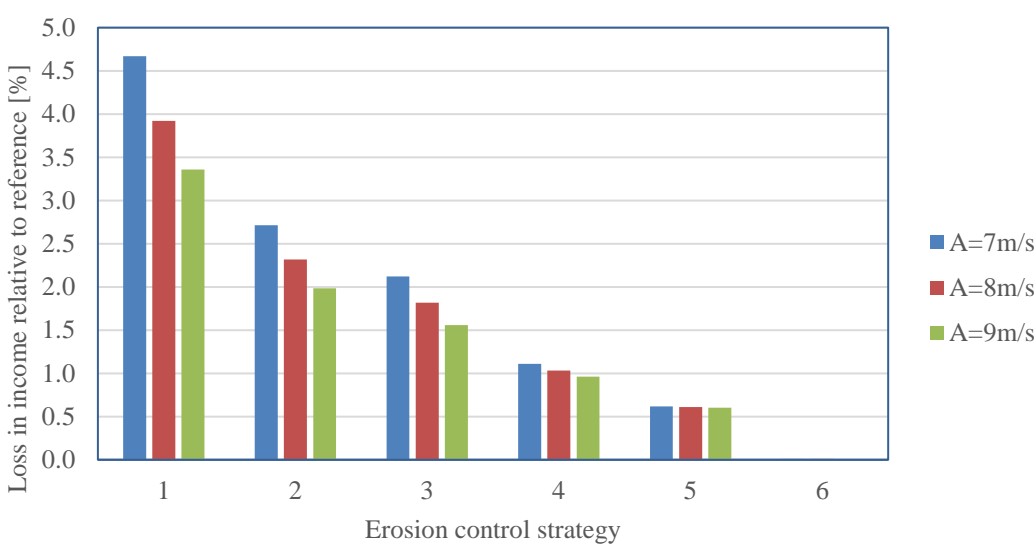

**Figure 20: Loss of income due to erosion, inspection and repair. Energy: 250€MWh]. Repair: 10000€/rotor. Inspection: 500€/rotor.**

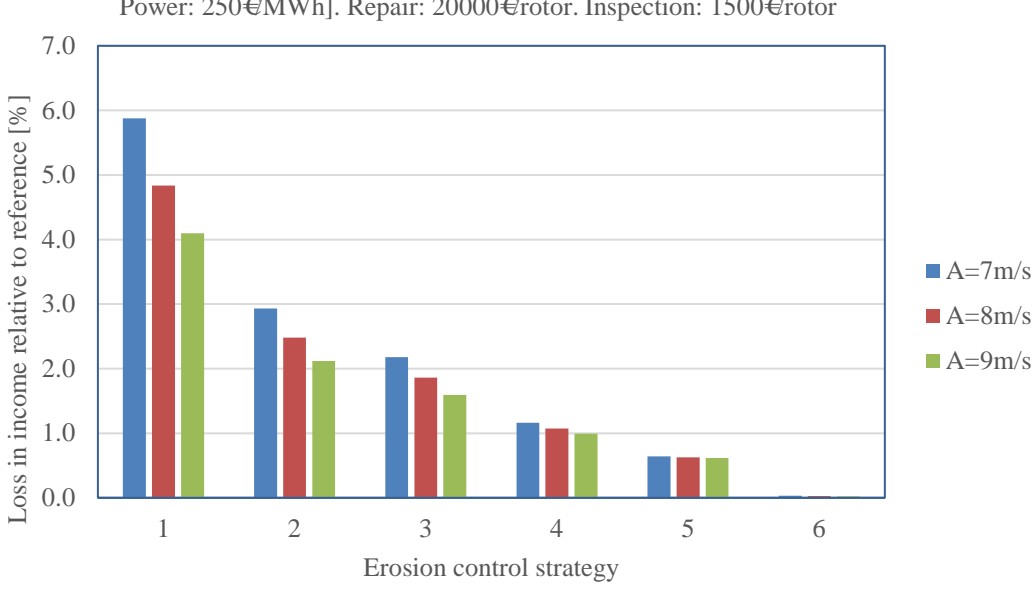

**Figure 21: Loss of income due to erosion, inspection and repair. Energy: 250€MWh]. Repair: 20000€/rotor. Inspection: 1500€/rotor.**

**6 Discussion**

This paper is a concept paper proposing a framework for prediction and mitigation of leading edge erosion. In order to demonstrate the concept quantitatively, a number of simplifying assumptions and approximations were made.

The assumption of homogenous droplet size for a given rain intensity is obviously an idealization of reality. For a given rain event the droplets sizes are distributed as explained in section 4. These correlations may vary a lot between different types of precipitation, climates, temperatures, levels of pollution, etc. Still the median droplet size and the frequency of large droplets generally increase with increasing rain intensity.

The assumption that the damage increment scales with the kinetic energy, and that the Wöhler curve for one droplet size can be extrapolated to other droplet sizes as suggested in section 3.2 may be controversial. However, there must be a strong correlation between the droplet diameter and the incremental damage. Very small droplets affect only the material very close to the surface. For surface cracking of brittle top coats the many impacts with smaller droplets may generate more accumulated damage than the few large droplets as demonstrated by Amirzadeh et al. (2017). For damage modes related to body waves propagating into the structure, affecting the material below the surface, like delamination and cracks in matrix, filler and top coats, only larger droplets and hail have sufficient kinetic energy and size of stress field to affect the structure below the surface. Thus the correlation between droplet size and damage increment depends a lot on the material, leading edge configuration and failure mode.

The assumption that the aerodynamic performance decreases linearly with time is not necessarily true. Typically, there will be a long incubation period, where the surface roughness is nearly unaffected, and then the roughness increases at a high rate.

The correlation between leading edge damage and loss of aerodynamic performance is not fully understood. The loss depends a lot on the aerodynamic profile of the blade and other factors. However, simulations and wind tunnel tests have been carried out, where leading edge roughness has been investigated and quantified. The transfer function between life time and aerodynamic performance is not understood.

The costs for inspection and repair also vary substantially. They are, however reported to be significant these years, in particular for off shore turbines.

The erosion issue has become significant, as the tip speed has increased along with the development towards larger turbines. Many modern turbines have tip speeds at the order of 80 to 90 m/s. In order to reduce the shaft torque in future designs, it may be attractive to increase tips speeds even beyond 100 m/s. Then occasional tip speed reduction for erosion control will be even more important, even when stronger leading edge designs are developed.

The expenses for establishing erosion control are not assessed. These will relate to control algorithms on the turbine control software, precipitation sensors in each wind turbine park and connections between the sensor and each turbine.

As demonstrated in section 5.2, the economic potential of erosion safe mode turbine control is significant. Even if the correlations between precipitation intensity and incremental damage or between degree of erosion and aerodynamic performance are not as strong as here assumed, the cost - benefit balance may still be in favor of erosion control.

## 7 Conclusions

A framework for prediction and a mitigation strategy for leading edge erosion was presented. The model takes into account the entire value chain: leading edge test data, actual on site precipitation, erosion rate, loss of production due to erosion, operation and maintenance. The lost energy production due to occasional tip speed reduction is marginal in proportion to the alternative of lost production due to eroded blades. Thus, the cost – benefit balance of erosion control looks very promising and shows a great potential for reducing the loss of produced energy due to erosion and the cost of operation and maintenance.

To accomplish erosion control there is a need for more knowledge on the correlation between precipitation and erosion for different leading edge structures and materials, and for development of methods and equipment for on-site now casting of precipitation.

Data availability. Please contact the corresponding author.

Author contributions. JIB had the lead on paper writing, test data analysis and lifetime prediction. CBH contributed mainly on precipitation and CB mainly on wind turbine control, aerodynamics and economy. All contributed to writing the paper.

Competing interests. The authors declare that they have no conflict of interest.

Acknowledgements.   The financial support from Innovation Fund Denmark (6154-00018B) for the project EROSION (www.rain-erosion.dk) is gratefully acknowledged. The authors want to acknowledge Polytech A/S for kindly providing the

laboratory test specimen images. Permission to use the GPCP annual mean precipitation (Fig. 9) is kindly granted by Todd Mitchell.

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
