# Peer review of "Extending the life of wind turbine blade leading edges by reducing the tip speed during extreme precipitation events"

_Wind Energy Science, 2017_

## Author Comment (AC2) · 26 Feb 2018

Please note, that manuscript for review and discussion was updated on Friday 23rd of February. The updated version has 31 pages (the old one had 35 pages).

The updates are mainly correcting some errors in numbering and cross references for tables and figures. I also adjusted the size of a few figures to eliminate the half-blank page and make the review version more reader-friendly.

With kind regards

Jakob

---

## Short Comment (SC1) · 28 Feb 2018

correction at page 22, line 20

current text: "In Figure to Figure 7 the loss of income due to lost AEP..."

will be corrected to: "In Fig. 16 to Fig. 19 the loss of income due to lost AEP..."

thank you

Jakob

---

## Referee Comment (RC1) · 14 Mar 2018

Wind power industry has been growing fast in the recent years. In my opinion, this manuscript has introduced a topic which is related to the journal scope. This manuscript discussed the topic in a good way and deserves to be published after some revisions:

1. The nomenclature is missing. Please add a complete nomenclature.

2. Line 23: "LEE is caused by a multitude of factors within the atmospheric environment and the leading-edge structure". Please Change this into "LEE is caused by a multitude

of factors within the atmospheric environment such as sand particles which intensively discussed in Zidane et al., 2017 and other airbone particles". Zidane, I. F., Saqr, K. M., Swadener, G., Ma, X., and Shehadeh, M. F.: Computational Fluid Dynamics Study of Dusty Air Flow over NACA 63415 Airfoil for Wind Turbine, Jurnal Teknologi, 70, 1-6, DOI: http://dx.doi.org/10.11113/jt.v79.11877, 2017.

3. More literature survey is to be added to present the most updated R&D status for further justification of the originality of the manuscript.

4. Please change the subtitle number 3.2 Analysis of rain erosion test data into 3.1 Analysis of rain erosion test data.

5. Line 23: Please provide an explanation why did you choose the falling velocity of the droplet to be 6 m/s.

6. Page 6 line 8: "as the damage progresses from an area of high velocity towards areas with lower velocity." Could you explain why this happens.

7. Page 7: Please figure 2, 3,4 and 5 should be 3, 4, 5 and 6.

8. Page 11 line 8: To account for variable conditions fatigue loading, different rules have been proposed for accumulation of damage in composites (Brøndsted et al. 1997). The most popular and easy to use, though not always correct, is the linear Palmgren-Miner rule. Could you explain how it is not always correct or the limitation of this rule? Also, how were you assured that it has worked in your presented study.

9. Page 17 line 7: "The higher the surface roughness, the thicker the boundary layer and the more reduction of the aerodynamic performance". It is better to replace this line with the following: "The drag coefficient of the airfoil increases, while the lift coefficient decreases especially at higher angles of attack causing severe losses in energy production".

10. Page 18 line 24: tables 3 to 7 not 4 to 8.

11. Please state that AEP refers to annual energy production in section 5.2 as it is not mentioned.

12. Adjust the figure numbering starting from figure 15 until the end.

13. Please merge the results and discussion sections.

14. Page 19 line 5: Please adjust the table numbers.

15. It is recommended to have one paragraph for the Conclusion.

---

## Referee Comment (RC2) · Anonymous Referee #2 · 20 Mar 2018

Summary of review: The manuscript presents relevant information for the wind industry, however, the methodology which is believed to be the core of the manuscript is not described. Reference and credit to previous work has not been made, this questions the originality of the manuscript and raises the question, what is the contribution of the authors to the scientific community? The manuscript can be accepted with major corrections. Specific comments about are provided next.

Specific comments: Section 1: Line 5-10 suggests that whirling arm test does not reflect real loading of blades as there are other environmental parameters that may impact its response. This is correct, but how is this linked to rain erosion testing?

Discussion on test factors affecting test data may be relevant.

Line 15-20 states the following: "Also, in order to reduce the torques and loads, it may be attractive to increase the tip speeds even further on future turbine designs. Consequently, alternative strategies of mitigation of LEE should be explored. Such an alternative strategy is the reduction of the tip speed during highly erosive conditions (Wobben, 2003). It is likely to be feasible to extend the leading edge life by reducing the rotor speed during extreme precipitation events occurring at a very little fraction of the service life, but accounting for the majority of the erosion damage." Please include definition of extreme event in terms of rain drop size and number of drops, or, state where it will be covered.

IEC61400-22 standard is a certification standard, how is this used in the context of the manuscript is unclear.

Section 2.2: Line 5 in this section sates: "Many designs have a layer of putty or filler 5 on the GFRP to make a smooth surface for the coating." Any reference on the statement above?

Section 3: No information about specimen geometry and material is provided, including thickness and roughness. Please include this information. This information is needed for the parameters presented in Table 1.

The following paragraph needs elaboration to make it more clear "It should be noted, that the time to removal of coating at position "i" is likely to be influenced by the adjacent erosion at position "i-1" as the damage progresses from an area of high velocity towards areas with lower velocity." Figure 3 reads ... Wöhler curve. Unclear what is the intention/definition to use or mention Wöhler curve. Further elaboration is needed.

Section 3.2: No references were provided and no description of units for each parameter. Is this the original contribution of the authors?

Section 4: This section does not define how rain data is used in the manuscript. A flow

chart will assist in the understanding of the methodology. Definition of convective rain is not provided. Please make sure all definition used in the manuscript are defined or provide reference where are those defined. Furthermore, Figure 8 refer to a reference, but not in the text, what is the author's intention here.

Section 5: Section 5.1.2: Reference to a model is made, however, model is not presented or defined. Please clarify and elaborate as needed. This will affect the whole section 5. Section 5.1.4 – Figure 12 has a far too long caption, please consider including the description into the body text of the manuscript. Section 5.1.5 – no references at all. Is this something the authors are proposing? Please update accordingly.

Section 6: The following is states "This paper is a concept paper proposing a framework for prediction and mitigation of leading edge erosion." The framework needs to clearly described. On page 21, the following is stated "The correlation between droplet size and damage increment depends a lot on the material, leading edge configuration and failure mode. For surface cracking of brittle top coats the many impacts with smaller droplets may generate more accumulated damage than the few large droplets as suggested by Amirzadeh et al., (2017). However, for elastomeric protective coating the damage mode may be debonding from the top coat/gelcoat, and in this case it may be opposite." Please elaborate and explain reason for this.

---

## Author Response (AR2)

Dear Jakob Mann.

Thank you very much for your comments and suggestions. Here are our answers:

Kind regards

Jakob Ilsted Bech.

1) There is no reason to have SI units in parenthesis in the equations. WES uses SI as default. I suggest removing these square brackets.

JAKB: removed

2) It is good to have a Nomenclature. However, it is not complete. I don't see DTV, rho, rho_l, rho_s, v_p^c etc. You have also abbreviations in the list but I don't see LER., AOA, GRFP, RET etc. Droplet diameter is "D" but "d" on p5 l2. Furthermore, in order to be useful it should appear in alphabetic order, Roman and Greek separately.

JAKB: it should now be complete

3) Do you have permission to show figure 1?

JAKB: yes.

4) Eq 3. Should rho be rho_l and c be c_l?

JAKB: yes, corrected

5) p4 l16 ")." missing after 1961.

JAKB: thank you

6) Eq 5: rho_p is not defined. Typo?

JAKB: yes. I changed it to s and l for solid and liquid as in eq. 2,3,4

7) Reference Cortés is incomplete.

JAKB: I added the DOI.

8) Table 1: The text describing the columns could be improved. Here are some suggestions: "Position of erosion front" instead of "Propagation of erosion", "Tangential speed at erosion front" instead of "Local tangential rotor speed", "Impact/…" is a bit hard to understand (same for axis label on figure 4). Is it number of impacts per square cm projected area at the position of the erosion front?? Text below table seems better placed in the main text or in the table caption.

JAKB: Thank you. I changed to your suggested text, plus a bit

9) Do you have permission to reproduce figure 7?

JAKB: yes.

10) Table 2 and Figure 9 add little compared to what is already in Figure 10. I think Table 2 and fig 9 should be dropped.

JAKB: I discussed it with Charlotte. Fig. 9 is based on detailed measurements but covers only latitudes between 40°N and 40°S. Fig. 10 covers from pole to pole but is based on less precise measurements and partly by modelling. We think, Table 2 shows additional content compared to the figures, and it gives some actual precipitation numbers from different references. We prefer to keep all three. I hope, you accept this

11) p15 l14: private communication with whom?

JAKB: private communication with Niels C. Therkildsen from Total Wind Group (blade repair). But we chose to use a written reference.

12) p18 l10: ECS4 appears twice.

JAKB: One is corrected to ECS5

13) p29 l5: "paragraph" -> "section".

JAKB: changed!